# The Role of *FT*/*TFL1* Clades and Their Hormonal Interactions to Modulate Plant Architecture and Flowering Time in Perennial Crops

**DOI:** 10.3390/plants14060923

**Published:** 2025-03-15

**Authors:** Lillian Magalhães Azevedo, Raphael Ricon de Oliveira, Antonio Chalfun-Junior

**Affiliations:** 1Laboratory of Plant Molecular Physiology, Plant Physiology Sector, Institute of Biology, Federal University of Lavras (UFLA), Lavras 37200-900, MG, Brazil; lillianmagalhaesazevedo@gmail.com (L.M.A.); rapharicon@gmail.com (R.R.d.O.); 2Department of Biological Sciences, State University of Santa Cruz (UESC), Ilhéus 45662-900, BA, Brazil

**Keywords:** plant development, *FLOWERING LOCUS T* (*FT*), *TERMINAL FLOWER 1* (*TFL1*), climate change, perennial plants, plant growth regulators

## Abstract

Human nutrition is inherently associated with the cultivation of vegetables, grains, and fruits, underscoring the critical need to understand and manipulate the balance between vegetative and reproductive development in plants. Despite the vast diversity within the plant kingdom, these developmental processes share conserved and interconnected pathways among angiosperms, predominantly involving age, vernalization, gibberellin, temperature, photoperiod, and autonomous pathways. These pathways interact with environmental cues and orchestrate the transition from vegetative growth to reproductive stages. Related to this, there are two key genes belonging to the same Phosphatidylethanolamine-binding proteins family (PEBP), the *FLOWERING LOCUS T* (*FT*) and *TERMINAL FLOWER 1* (*TFL1*), which activate and repress the floral initiation, respectively, in different plant species. They compete for transcription factors such as *FLOWERING LOCUS D* (*FD*) and 14-3-3 to form floral activation complexes (FAC) and floral repression complexes (FRC). The *FT*/*TFL1* mechanism plays a pivotal role in meristem differentiation, determining developmental outcomes as determinate or indeterminate. This review aims to explore the roles of FT and TFL1 in plant architecture and floral induction of annual and perennial species, together with their interactions with plant hormones. In this context, we propose that plant development can be modulated by the response of *FT* and/or *TFL1* to plant growth regulators (PGRs), which emerge as potential tools for mitigating the adverse effects of environmental changes on plant reproductive processes. Thus, understanding these mechanisms is crucial to address the challenges of agricultural practices, especially in the face of climate change.

## 1. Introduction

Flowering is crucial in the reproductive cycle of plants and exerts a direct influence on their survival and perpetuation. Furthermore, in crops of economic interest, flowering timing plays a fundamental role in determining productivity and harvest yield [1,2]. Climate change poses a substantial challenge to agricultural production, particularly in terms of affecting the flowering period and subsequent yield [3,4]. Extreme weather events, such as heat waves and droughts, can disrupt the critical flowering stage, leading to reduced production [5]. For example, thermal stress during flowering has a more significant negative effect on maize, an annual crop, than on other stages of the growing season, with temperatures exceeding 30 °C causing substantial yield losses [5].

Similarly, coffee plants, an important perennial crop, are highly vulnerable to climate change. Studies indicate that an increase in temperature and changes in precipitation patterns due to climate change can lead to the abortion of coffee flowers and premature ripening of coffee beans, ultimately reducing the area suitable for coffee cultivation [6,7,8]. Climate models predict a decrease in coffee production by approximately 28% in Latin America and 12% in Africa, highlighting the detrimental effects on production [9,10]. Additionally, the incidence of pests and diseases is expected to increase as a result of climate change, further compromising coffee production and quality [11]. Projections also suggest an increase in temperature variability, precipitation, and soil moisture, which will affect the reliability of the flowering period and complicate management practices such as fertilization and irrigation [12]. 

Changes in climate patterns, including rising temperatures and irregular precipitation, necessitate adaptive measures to mitigate adverse impacts on agricultural production and ensure food security [3,4,13,14,15]. As a result, numerous studies have focused on understanding the factors triggering the transition from the vegetative to the reproductive phase. Molecular analyses have provided significant insights into how plants integrate endogenous and exogenous signals to regulate flowering, with special attention to gibberellins (GA), age-dependent factors, temperature, photoperiod, vernalization, and autonomous pathways [16,17,18]. 

During floral induction, the expression of many genes is regulated in response to different environmental signals [19]. In this context, *FLOWERING LOCUS T* (*FT*) and *TERMINAL FLOWER 1* (*TFL1*) genes, which belong to the phosphatidylethanolamine (PEBP) family of binding proteins, are known for their roles in the induction and repression of flowering, respectively [20]. These genes play essential and antagonistic roles in controlling shoot meristem identity, flowering time, and plant architecture [21,22,23,24]. 

Plant architecture is shaped by stem organs that originate from the dynamic and particular cell division and differentiation of the shoot apical meristem (SAM) and the axillary meristems originating from it in each species. However, after floral transition, SAM or axillary meristems take the form of an inflorescence meristem (IM), which can develop into a floral meristem (FM), thus determining growth as a reproductive phase or maintaining it as a vegetative phase [25]. This process is influenced by the activities of *FT* and *TFL1* genes, in which TFL1 maintains the indeterminate meristem, resulting in the formation of leaves and branches, whereas some member of the *FT* family converts the indeterminate meristem into determinate, triggering the formation of flowers [26,27,28].

These genes, notably, show a high degree of conservation, with the encoded proteins sharing similarities in amino acid sequence [29]. Thus, FT and TFL1 compete for binding with transcription factors such as bZIP, FLOWERING LOCUS D (FD), and the 14-3-3 protein, respectively forming the floral activation complex (FAC) or the floral repression complex (FRC) in the SAM [30,31,32]. These molecular interactions play a fundamental role in determining the fate of the meristem and, therefore, the final architecture of the plant, directly connecting gene expression with morphological development [28].

The *FT* has emerged as one of the most widely studied and characterized genes, playing a crucial role in inducing flowering [33]. Its expression is activated in response to a series of internal and external signals, especially in the leaves. The FT protein can be translocated through the phloem to the SAM. Additionally, *FT* mRNA, along with the FT protein, can move systemically and function as an integral component of the florigenic signal that induces flowering [34,35,36]. In SAM, FT interacts with the transcription factor FD, triggering a cascade of events that culminate in the expression of genes such as *SUPPRESSOR OF OVEREXPRESSION OF CONSTANS 1* (*SOC1*), resulting in the transcription of *APETALA 1* (*AP1*) and *LEAFY* (*LFY*), fundamental for the identity of the floral meristem and the formation of floral organs [37,38]. In contrast, an antagonistic relationship was observed between *TFL1* and genes that determine floral identity. Overexpression of *TFL1* in IM regulates the expression of *AP1* and *LFY*, whereas *AP1* and *LFY* repress *TFL1* expression in FM [39]. Furthermore, MADS-box transcription factors, such as *SHORT VEGETATIVE PHASE* (*SVP*), *SUPPRESSOR OF OVEREXPRESSION OF CONSTANS 1* (*SOC1*), *AGAMOUS-LIKE 24* (*AGL24*), and *SEPALLATA 4* (*SEP4*) also suppress *TFL1* and are regulated by *AP1* in FM. However, it is important to note that appropriate levels of expression of these genes are necessary for *AP1* to exert its function of suppressing *TFL1* in FM [25,40].

In this context, the coordinated regulation of *FT* and *TFL1* can influence plant architecture and life cycle but also plays a significant role in determining the timing of flowering. Understanding how these genes interact in SAM and respond to climate change is fundamental to unraveling the mechanisms that govern the transition from the vegetative to the reproductive phase. Moreover, it is necessary to develop strategies for mitigating the adverse effects in future scenarios. For example, the use of plant growth regulators (PGRs) can be a viable strategy, as they can modulate plant growth and development to regulate flowering and the subsequent formation and maturation of fruits, particularly in crops. Additionally, climate change can affect production and, consequently, human nutrition [41]. These strategies can help improve management practices for food production, ensuring better adaptation and resilience in the face of environmental challenges.

## 2. The *FT*/*TFL1* Developmental Regulatory Pathways: From Model Species to Perennial Crops

In the plant kingdom, reproductive patterns vary between two main categories: annual and perennial. Although annual species complete their entire life cycle in a single year, perennial species generally take longer to enter their reproductive state, sometimes requiring several years [42].

In studies focused on annual species, especially *Arabidopsis*, the balance between FT/TFL1 has been the subject of investigation, directly associating it with the fate of the meristem [22,43]. The dynamics of this balance are strongly influenced by the plant’s age. In young plants, the concentration of TFL1 at the apex is high relative to that of FT, resulting in a low FT/TFL1 ratio, which maintains the meristem in a vegetative state. However, as the plant grows and more leaves are formed, a larger pool of FT accumulates in the SAM due to the transport of FT from the leaves [31]. This increase in the FT/TFL1 ratio triggers the transition from the vegetative to reproductive meristem, allowing the plant’s life cycle to be completed [31]. Thus, the moderate relationship between FT/TFL1 allows balanced development between branches and flowers [24].

Understanding the mechanisms of *FT*/*TFL1* gene regulation is not only limited to annual plants but also extends to perennial species, where the interaction between these genes plays a crucial role in controlling the plant life cycle. In contrast to annual plants, perennial plants exhibit high levels of TFL1 in the SAM, maintaining it in a vegetative state, whereas in the axillary meristems, the level of FT is higher, activating genes related to floral meristem identity [31]. Downregulation of *TFL1* accelerates flowering in several perennial species, such as poplar, apple, and pear [44,45,46], highlighting the importance of TFL1 in these processes. Furthermore, expression of the *FT* and *TFL1* orthologs, *SINGLE FLOWER TRUSS* (*SFT*) and *AUTO-PODA* (*SP*), respectively, directly influence production in cultivated plants such as the tomato [24]. 

More recently, it was discovered that *TFL1* is also expressed in the leaves and mobile in tobacco and tomatoes [47]. Thus, the movement of PEBP may be a conserved mechanism across different plant species, indicating that both FT and TFL1 can take the same pathway, signaling whether the plant should produce flowers [47]. This recent finding suggests that flowering regulation involves a more complex interaction between FT and TFL1 than previously understood. The presence of *TFL1* in the leaves indicates that both proteins could be simultaneously translocated to the SAM, where their relative concentrations and interactions would influence the decision to activate floral development. In perennial plants, this dual expression could allow for more flexible and responsive control of flowering, enabling plants to adapt to different environmental conditions and maintain continuous or multiple flowering cycles throughout the year [48,49,50]. This finding implies that the mechanisms governing flowering are more complex, potentially involving additional regulatory layers and interactions between *FT* and *TFL1*. For example, miRNAs target *FT* and *TFL1* in various species, including perennial plants [51,52,53]. These non-coding RNAs affect gene expression by influencing the mRNA cleavage, translation, or DNA methylation of target genes [54]. In coffee plants, miRNAs involved in microsporogenesis, the regulation of hormonal responses, and the modulation of transcription factors associated with flowering in response to environmental changes have been identified [52,53]. Additionally, in *Dendrobium catenatum*, miR156 regulates *FT* and *TFL1* expression during different developmental stages, including the juvenile–adult transition and plant maturation [51]. This highlights the need for further research to understand how the spatial and temporal expression patterns of these genes affect flowering, which could have significant implications for agricultural practices, especially in the context of climate change and the cultivation of perennial crops.

The role of these genes has been studied in various perennial crops in response to seasonal variations throughout the year to understand how their regulation influences flowering and, consequently, fruit production [43,49,50,55]. For example, Cardon et al. [50] demonstrated that *CaFT1* acts as a florigen ortholog, exhibiting active transcription from February to October across various *Coffea* spp. genotypes. This suggests a continuum of floral induction that offers multiple starting points for floral activation, thereby explaining the asynchrony and prolonged flowering events observed in the perennial tropical species. Similarly, it has been demonstrated that hops (*Humulus lupulus* L.) can flower multiple times throughout the year under subtropical conditions in Brazil, regardless of the season, due to consistently inductive photoperiods [55]. Additionally, the study suggests that under these conditions, the expression of *TFL* may not be sufficient to induce branching and inhibit *HlFT* in hops, resulting in multiple flowering events [55].

However, divergences in the literature arise when exploring the FT/TFL1 relationship between annual and perennial plants. Although the antagonistic role of FT and TFL1 can be explained by their competitive interaction with FD, studies with transgenic *Arabidopsis* have revealed a greater affinity of FT with FD to form a floral activation complex (FAC) [43]. This dynamic suggests the possibility of a dominant role for FT in counterbalancing the repressive effect of TFL1 before floral transition [56]. In contrast, in perennial species such as pear (*Pyrus pyrifolia*), studies indicate that *PpFT1* expression is not upregulated during floral induction but that a reduction in *PpTFL1* expression is essential to induce flowering [57]. Furthermore, ectopic overexpression of *ScTFL1* in *Arabidopsis* delayed flowering unexpectedly; overexpression of florigen-like *ScFT1* also caused delayed flowering. This suggests that the roles and expression patterns of *FT* and *TFL1* gene family members in *Saccharum* spp. diverged from those of similar PEBP family members [58].

These contrasting results can be attributed to the evolutionary history of PEBP genes and the species, which present differences in life cycles such as development and competition strategies resulting in annual and perennial plants. Annual plants complete their life cycle in a single growing season, requiring a rapid and decisive transition to flowering, which may necessitate a stronger and more straightforward FT-FD interaction to ensure reproductive success [43,56,59]. In contrast, perennial plants, which have lived for several years, have developed more complex regulatory mechanisms to balance vegetative growth and flowering over extended periods [49,60]. For instance, the expression of *TFL1* in leaves and its mobility can provide perennials with an adaptable system to respond to various environmental conditions, promoting asynchronous and prolonged flowering cycles [26,49,61]. This indicates that in perennials, the regulation of flowering is not solely dependent on simple FT/TFL1 antagonism but involves a complex interaction of spatial and temporal expression patterns, as well as additional environmental factors. Therefore, by manipulating the expression of these genes or using plant growth regulators, it may be possible to optimize flowering time and improve fruit production in perennial crops, thereby enhancing agricultural resilience.

## 3. Interaction Between *TFL1* and Plant Hormones

Most studies have focused on the interactions between plant hormones and *FT*, a flowering inducer [62]. However, the interactions between *TFL1* and these hormones remain largely unexplored. Therefore, it is essential to understand not only the individual roles of FT and TFL1 but also the balance between them in response to hormonal signals, particularly in perennial species, where research on this topic remains limited. Therefore, this topic aims to review the existing literature regarding the regulation of *TFL1* expression by plant hormones.

In addition to the known roles of FT and TFL1 in determining plant architecture, the interactions between these genes and hormones play a significant role in this process. Auxin, which is mainly produced in the SAM, plays a fundamental role in maintaining apical dominance and stem elongation, inhibiting the growth of lateral branches, whereas cytokinin (CK) promotes cell division and stimulates the growth of lateral branches [63]. The balanced regulation of these hormones is essential for the control of vegetative growth and, consequently, reproductive growth [64], which has prompted investigations into the interaction between FT/TFL1 and hormones in shaping plant architecture and promoting flowering [57,65,66,67,68].

In studies carried out in apples (*Malus domestica*), it was observed that the combination of auxin and cytokinin in SAM increased the expression of *MdTFL1*, maintaining vegetative growth [66]. Auxin in the SAM promotes stem elongation, maintains apical dominance, and inhibits lateral bud growth. Cytokinin promotes cell division and works together with auxin to induce the expression of *MdTFL1* in the shoot apex. Conversely, during the transition to the reproductive phase, the decrease in auxin levels favors the expression of *MdFT*, determining the fate of the shoot apical meristem [66]. Furthermore, the exogenous application of 6-BA (6-benzylaminopurine), a CK that stimulates floral bud formation in apple trees and is applied before floral induction, has an indirect effect on the endogenous ratio between zeatin (ZT, an active tZ-type cytokinin) and IAA [69,70]. The increase in zeatin and the decrease in IAA due to 6-BA treatment may help disrupt the hormonal balance that typically suppresses flowering [69,70]. This study suggests that these hormonal changes contribute to the downregulation of *MdTFL1* and, consequently, to the increased expression of *AFL1* (*APPLE FLORICAULA*/*LFY*) at the onset of flowering, thereby promoting floral development [69,70]. In pear (*Pyrus pyrifolia*), some genes related to the biosynthesis and signaling of hormones, such as auxin, CK, abscisic acid, and ethylene, showed a synergistic or antagonistic co-expression pattern with *PpTFL1* during floral induction [57]. While the expression of genes related to auxin and abscisic acid tended to decrease, the expression of genes related to CK and ethylene pathways increased during this process [57]. These results suggest that in perennial plants, the hormones auxin, abscisic acid, and gibberellin can maintain *TFL1* levels higher than *FT*, while ethylene and cytokinin tend to increase *FT* expression (Figure 1). In agreement, transcriptomic analyses in coffee identified 12.478 differentially expressed genes in the shoot apical meristem, floral buds, and leaves, with enriched terms primarily related to hormonal regulation during the flowering induction phase. These findings suggest that hormones play a central role in triggering flowering and may directly or indirectly regulate *FT* and *TFL1* genes [48,71].

Additionally, abscisic acid (ABA) plays a role in delaying floral transition in Arabidopsis, positively influencing the expression of *FLOWERING LOCUS C* (*FLC*) through the transcription factor ABI5 [72]. Studies have revealed that ABA functions as a negative regulator of branching, potentially suppressing elements of the cell cycle machinery as well as indole-3-acetic acid (IAA) biosynthesis and transport [73]. These findings suggest a complex interaction among ABA, ABI5, and TFL1, which potentially affects the expression of other genes involved in floral transition and branching. Furthermore, ethylene also influences the expression of *TERMINAL FLOWER 1* (*TFL1*), as demonstrated by studies showing that the ethylene receptor ETR2 can delay floral transition and affect starch accumulation in rice, possibly through downstream gene regulation, such as *TFL1* [74]. These connections reveal an intricate network of molecular regulation between PEBP family members and plant hormones that coordinate floral transition and development in plants.

Some studies have shown that the application of gibberellin (GA) upregulates the *TFL1* homolog and inhibits flower formation in perennial species, such as apples and roses [65,68,75]. In saffron (*Crocus sativus* L.), it was suggested that DELLA negatively regulates *TFL1*, showing its role in the GA signaling pathway [64]. The same was reported for grapevine [76]. Therefore, in perennial plants that exhibit repression of flowering by GA, TFL1 may be a key point of this repression and may be a direct target of GA-responsive transcription factors (Figure 1) [68,77]. These studies demonstrate that the use of plant growth regulators (PGRs) can alter the expression of *FT* and *TFL1* genes, potentially influencing the flowering time of these plants and consequently regulating their architecture [48,71]. 

Although these studies have expanded our understanding of the interplay between FT/TFL1 and plant hormones in regulating flowering, it is important to recognize that significant gaps remain, and many assumptions remain to be clarified. A schematic representation of floral regulation in perennial plants by PGRs is shown in Figure 1, illustrating a possible pattern of co-expression between plant hormones and *FT* and *TFL1* genes based on the results reviewed in this section. This knowledge can be expanded for the practical applications of these PGRs in the field, enabling the regulation of flowering time and synchronization.

## 4. Conclusions

Flowering regulation in plants, governed by genes such as *FLOWERING LOCUS T* (*FT*) and *TERMINAL FLOWER 1* (*TFL1*), is a pivotal aspect that influences both plant architecture and agricultural productivity. As climate change intensifies, understanding these regulatory mechanisms has become increasingly critical for sustaining global food security. This review underscores the conserved pathways involving *FT* and *TFL1* across annual and perennial species, highlighting their interactions with plant hormones and environmental cues. These interactions not only dictate the transition from vegetative to reproductive phases but also determine the adaptability of plants to changing environmental conditions.

The intricate balance between FT and TFL1, mediated by hormonal signals, such as auxins, cytokinins, gibberellins, ethylene, and abscisic acid, underscores the complexity of flowering regulation. In annuals, the timely expression of *FT* triggers floral transition, whereas in perennials, TFL1 maintains meristematic indeterminacy, enabling continuous or periodic flowering responses to environmental cues. These insights suggest promising avenues for leveraging plant growth regulators (PGRs) to optimize flowering times. Such efforts not only promise to sustainably improve agricultural practices but also provide strategic tools to mitigate the impacts of climate change on agricultural production.

Although significant advances have been made in understanding the relationship between FT and TFL1 and their interaction with plant hormones, several questions remain unanswered. Comparative studies among perennial species can help identify conserved and divergent patterns in the regulation of these genes, especially in tropical and subtropical crops, where the molecular regulation of flowering under variable environmental conditions is still poorly explored.

Another fundamental aspect to be investigated is the interaction of *FT*/*TFL1* with hormonal networks in different environmental contexts. There are still gaps in understanding how plant hormones regulate *TFL1* expression. To address this, large-scale transcriptomic and metabolomic approaches can provide new information on these interactions and their responses to seasonal changes and environmental stresses.

Furthermore, the discovery of TFL1 protein mobility raises new questions about its impact on plant architecture. Detailed functional studies using genetic editing and molecular tracking can provide information on how PGR-mediated signaling regulates *FT*/*TFL1* expression and, consequently, influences the balance between vegetative and reproductive growth in woody and perennial species.

## Figures and Tables

**Figure 1 plants-14-00923-f001:**
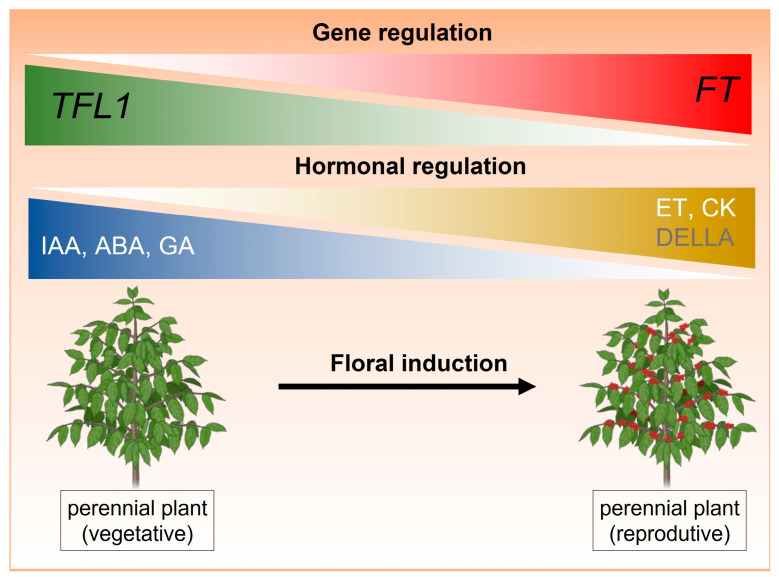
Proposed model of co-expression between *FT* and *TFL1* with plant hormones during the transition from the vegetative to reproductive phase in perennial plants. Gene abbreviations (black text within boxes) represent key regulatory genes, while hormone abbreviations (white text) indicate plant hormones involved in this process. DELLA, a regulator of gibberellin (GA) signaling, is represented in gray. *FT*: *FLOWERING LOCUS T*, *TFL1*: *TERMINAL FLOWER 1*, IAA: auxin, ABA: abscisic acid, GA: Gibberellin, ET: ethylene, CK: cytokinin.

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
