# Peer review of "The Role of FT/TFL1 Clades and Their Hormonal Interactions to Modulate Plant Architecture and Flowering Time in Perennial Crops"

_plants, 2025, doi:10.3390/plants14060923_

Round 1

Reviewer 1 Report

Comments and Suggestions for Authors

Reviewer 2 Report

Comments and Suggestions for Authors

The transition of both annual and perennial plants from vegetative growth to the reproductive stage is of great importance, as it largely determines the yield of plants. This transition depends on many exogenous and endogenous factors. Among the endogenous factors, phytohormones play an important role. The mechanism of flowering transition is not well understood, but it is known that two proteins FLOWERING LOCUS T (FT) and TERMINAL FLOWER 1 (TFL1) activate and repress flowering initiation, respectively, and play a central regulatory role in different plant species. The FT/TFL1 ratio largely determines the transition to the flowering stage. Phytohormones can have a significant effect on the FT/TFL1 ratio and thus on the plant's transition to flowering.

I think the review is well written but peculiar. Usually when writing a review, authors try to analyze all available literature in as much detail as possible. It seems to me that the authors of this review did not set themselves such a task (though I do not deal with the problem of flowering and may be wrong). It seems that they looked at the problem from a high level in order to emphasize the main points and not to get bogged down in parsing the details. If they were trying to describe everything in detail, it would be logical to give some including molecular information characterizing FT and TFL1. Very few results are given on the effects of phytohormones on the genes encoding FT and TFL1. If one were to even analyze the available transcriptome data, one would probably find many interesting results.

What and how the authors of the paper intended to write, I don't know, but they have produced a short review without much detail, useful to introduce the reader to the problem. For such a purpose it is very suitable.

There are small mostly technical remarks, which are given without numbering, but mostly with the lines to which they refer.

It is necessary to check the correctness of the spelling of the names of genes and protein -straight type or italics (lines 81, 86, etc.)

140-141 “Thus, the movement of PEBP mRNA may be a conserved mechanism across different plant species” and lines 144-145 “...both proteins could be simultaneously translocated to the SAM...”. Which moves mRNA or proteins after all?

160 The word “expression” is better used with the word “gene” rather than “protein”.

218-219 “...combination of auxin and cytokinin...” in what sense is the word “combination” used here?

222-224 “The exogenous application of 6-BA (6-benzylaminopurine)... has an indirect effect on the endogenous ratio between Zeatin and IAA”.  How can 6-BA affect the ratio of IAA to Zeatin? As you know, there are several forms of zeatin. What kind of zeatin are we talking about?

Will it not appear to readers that in Figure 1 in the combination of “ET, CR and DELLA”, the authors attribute “DELLA” to phytohormones.

In this short article the authors talk about food security 3 times.This is too much for a scientific article.

Reviewer 3 Report

Comments and Suggestions for Authors

The manuscript is well-written and of high research interest. Flowering time is a crucial trait, and this revision will have a significant impact on the scientific community. Given the importance of phytohormones, a more in-depth review of this aspect is essential to strengthen the manuscript. Numerous hormonal genes, such as those related to ethylene, have been implicated in the regulation of flowering time and should be incorporated into the revision.
